# Volatiles of All Native *Juniperus* Species Growing in Greece—Antimicrobial Properties

**DOI:** 10.3390/foods12183506

**Published:** 2023-09-20

**Authors:** Evgenia Fotiadou, Evgenia Panou, Konstantia Graikou, Fanourios-Nikolaos Sakellarakis, Ioanna Chinou

**Affiliations:** 1Laboratory of Pharmacognosy and Chemistry of Natural Products, Faculty of Pharmacy, National and Kapodistrian University of Athens, Panepistimiopolis, Zografou, 15771 Athens, Greece; eugenia.fot@gmail.com (E.F.); evgenia.panou39@gmail.com (E.P.); kgraikou@pharm.uoa.gr (K.G.); 2Society for the Protection of Prespa, Agios Germanos, 53077 Florina, Greece; fansakell@gmail.com

**Keywords:** greek juniper species, cupressaceae, cones, leaves, volatiles, GC-MS, antimicrobial activity

## Abstract

Juniper (*Juniperus* L., Cupressaceae Bartlett) trees are of high commercial value, as their essential oils are widely applied in the food and cosmetic industries due to their bioactivities. The genus *Juniperus* comprises eight species in Greece, and in the current work, we report the chemical analyses of their volatiles (GC-MS) obtained from the leaves and cones of all indigenous species found in the country, as well as their antimicrobial properties. The studied species were *J. oxycedrus* L., *J. excelsa* M. Bieb., *J. foetidissima* Willd., *J. communis* L., *J. macrocarpa* Sibth. & Sm., *J. turbinata* Guss., *J. sabina* L. and *J. drupacea* Labill., and a total of 164 constituents were identified. Monoterpenes, followed by sesquiterpenes, appeared as the dominant compounds in all investigated species. Most of the studied essential oils belonged to the α-pinene chemotype, with amounts of α-cedrol, sabinene, limonene and myrcene among the abundant metabolites, except for *J. sabina*, which belonged to the sabinene chemotype. Through antimicrobial tests, it was observed that the essential oils of most of the cones showed better activity compared with the respective leaves. The essential oils of the cones of *J. foetidissima*, *J. communis* and *J. turbinata* showed the strongest activity against the tested microorganisms. Additionally, in these three species, the content of thujone, which is a toxic metabolite found in essential oils of many *Juniperus* species, was considerably low. Taking into consideration the chemical profile, safety and antimicrobial activity, these three Greek *Juniperus* species seemed to provide the most promising essential oils for further exploitation in the food and cosmetics industries.

## 1. Introduction

The genus *Juniperus* L. (Cupressaceae Bartlett, class of the Gymnosperma) includes approximately 75 species of evergreen perennial coniferous shrubs or small trees [1]. Species of the genus have been widely used since ancient times, either as food or spice or for their medical and therapeutic properties [2]. Several spirits flavored with juniper, together with other botanical ingredients, are consumed worldwide and have a very important impact on the market [3]. In particular, *J. communis* L. has the largest range of any woody plant in the cool temperate geographical locations of the Northern Hemisphere and the essential oil of its cones is widely used as a traditional medicine and by the pharmaceutical industry, with applications in the food industry as a spice and meat preservative and in the beverage (as flavoring) industry [4]. Moreover, the burning branches and leaves (needles) of the plant produce smoke that has been used for smoking meat, cooked meat and fish since antiquity [5]. The most famous juniper-flavored alcoholic beverage is gin, which is a spirit with a rich cultural history over the centuries and has been prepared as a medical treatment since the 11th century, rising in popularity between the 14th and 15th centuries up until now, where it is in wide use and consumption as a beverage [3,6]. The name of gin is a shortened form of the English word genever (from French genièvre and Dutch jenever), keeping its etymology from the Latin name of juniper, namely, “juniperus” [7].

As a medicinal herb, juniper had been an essential tool of doctors’ remedies for centuries: the Romans were burning juniper branches for purification, and plague doctors stuffed their characteristic plague masks with juniper in order to protect themselves from the Black Death. Across Europe, pharmacists have used tonic wines based on juniper to treat cough, cold and pains [5,8].

Due to the well-known medicinal properties, there are two monographs in the European Medicines Agency (EMA) for both *J. communis* dried cones and its essential oil for their traditional use as a diuretic and the relief of digestive, gastrointestinal and/or musculoskeletal disorders [9,10]. The Juniperus plants are also mentioned in several national Pharmacopoeias (such as the German Pharmacopoeia (DAB 10), Austrian Pharmacopoeia (ÖAB 90), French Pharmacopoeia (Ph. Fr. X), Swiss Pharmacopoeia (Ph. Helv.), British Herbal Pharmacopoeia (BHP) and European Pharmacopoeia (Ph. Eur.)) as medicinal plants. In particular, in the Ph. Eur. 6.0, the herbal material is described as the “dried ripe cone berry of *J. communis* L.” and was named *Juniperus* pseudo-fructus. In the most recent edition of the European Pharmacopoeia the plant part was described as *Juniperi galbulus* [10]. Furthermore, *Juniperus* essential oils are widely applied in the cosmetics industry, mainly for nourishment, protection and repair of skin [11,12].

According to the literature, the chemistry of the *Juniperus* genus has been broadly studied; junipers are characterized by a rich chemical profile containing a high percentage of terpenes, flavonoids, phenolic acids, resin, tannins and organic acids [4,13,14]. Due to these metabolites, but particularly due to the production of essential oils, junipers exert antimicrobial [15,16,17], antioxidant [15,16,18,19], anti-inflammatory [20,21], antidiabetic [22,23], hepatoprotective [18,24], insecticidal activity [25] and cytotoxic [26,27,28,29] properties.

Essential oils of junipers are aromatic and volatile liquids that accumulate in the cones and leaves. Two terpene chemical groups, namely, monoterpenes and sesquiterpenes, constitute the main part of juniper essential oils and produce their strong and distinctive aroma, as well as their antibacterial and antifungal properties [14]. The most commonly predominant essential oils’ constituent is the bicyclic monoterpene α-pinene, which exerts the characteristic pine tree flavor [30].

According to [31,32], the genus *Juniperus* is represented by eight species in the flora of Greece, viz., *J. communis* L., *J. drupacea* Labill., *J. excelsa* M.Bieb., *J. foetidissima* Willd., *J. macrocarpa* Sibth. & Sm., *J. oxycedrus* L., *J. sabina* L. and *J. turbinata* Guss. With respect to infraspecific taxa, *J. communis* is represented in Greece by *J. communis* subsp. *communis*, while *J. oxycedrus* is represented by *J. oxycedrus* subsp. *deltoides* (R.P. Adams) N.G. Passal. It has to be noted that until recently, *J. turbinata* was regarded as a synonym of *J. phoenicea* subsp. *turbinata* L.; nevertheless, and according to [33], the latter is confined to the Mediterranean parts of Spain and France, while the former has a wider distribution ranging from the Canary Islands and Madeira to the Sinai Peninsula.

In the current study, the chemical profiles of the essential oils of the eight *Juniperus* species growing wild in Greece were analyzed and compared with previous literature data. The plant material was collected from different geographic areas of Greece (Table 1), with four of them taken from Prespa Lakes National Park in the framework of our studies on selected plants and bee-keeping products from this unique area [34]. Lake Prespa (NW Greece) is a tectonic lake in the Balkan Peninsula with a rich and unique biodiversity in flora (>1800 plant taxa) and fauna, and one of the smallest areas in the Balkan Peninsula with big numbers of Greek juniper trees (*J. excelsa*) [35]. The tree-shaped Greek juniper (*J. excelsa*) and stinking juniper (*J. foetidissima*) appear to be mostly rare and scattered. Prickly juniper (*J. oxycedrus*) is represented in the region throughout the Southern Balkans and the Eastern Mediterranean by subsp. *Deltoides*, which is a shrub rarely exceeding 2.5 m. Common juniper (*J. communis* subsp. *communis*) is pervasive in the national park only on the rough pasture lands that are widespread above 1400 m, where any beech and fir forests were cleared centuries ago. Further up, *J. communis* subsp. *communis* characterizes the high montane and subalpine zone of Mount Varnous. Woodlands of the tree junipers *J. excelsa* and *J. foetidissima* are not common in Southern Europe. They are western outposts of the far more extensive juniper woodlands in southern Anatolia and the Near East. For the people of the Prespa region in Greece, the juniper woods are an integral part of their cultural history and were of everyday importance to life in the past. Residents used the extremely durable juniper wood for building houses, fences, roofs and boats, as well as fish traps. Moreover, they were used as spices and for medical purposes [35].

Among the studied species, the essential oils of *J. macrocarpa*, *J. sabina* and *J. excelsa* cones, as well as *J. communis* subsp. *communis* leaves, were studied, to our knowledge, for the first time. In addition, all samples were evaluated for their antimicrobial activity.

## 2. Materials and Methods

### 2.1. Plant Material

Leaves and cones of juniper shrubs and trees were collected as described in Table 1 and Appendix A. The average temperature during the period of plant collection was between 16 and 20 °C. The plant samples were naturally dried at 25 °C in a well-ventilated environment for about 10 days. After the drying process, the cones and leaves were pulverized and stored in darkness at 25 °C.

### 2.2. Isolation of Essential Oils

Quantities of aerial parts, leaves and cones were hydro-distilled on a Clevenger-type apparatus for 3 h at 100 °C. Each essential oil, after adding anhydrous sodium sulfate, was stored in a dark glass container at approximately 4 °C.

### 2.3. GC-MS Analysis

The chemical compositions of the essential oils were analyzed using GC-MS. The analysis was performed on an Agilent Technologies Gas Chromatograph 7820A connected to an Agilent Technologies 5977B mass spectrometer system (Agilent, Santa Clara, CA, USA) based on electron impact (EI) and 70 eV of ionization energy. The gas chromatograph was equipped with a split/splitless injector and a capillary column HP5MS of 30 m, internal diameter of 0.25 mm and membrane thickness of 0.25 μm. The temperature program included an initial temperature of 60 °C for 5 min; then, an increase at a rate of 3 °C/min until the temperature reached 130 °C; then, an increase at a rate of 2 °C/min up to 180 °C; and finally, an increase at a rate of 5 °C/min with a final temperature of 240 °C, where the program was completed. The total analysis time was 65.33 min. The carrier gas was He at a flow rate of 0.7 mL/min, injection volume of 2 μL, split ratio of 1:10 and injector temperature of 280 °C. The compounds were identified via mass spectra comparison with libraries (WileyRegistry of Mass Spectral Data) and confirmed via a comparison of Kovats retention indices (KIs) with literature data [36].

### 2.4. Antimicrobial Activity

All essential oils were dissolved in dimethyl sulfoxide (DMSO) using Mueller–Hinton broth or RPMI with MOPS for the growth of fungi, and were further investigated for their antimicrobial activity against eight (8) strains of human pathogenic bacteria and three (3) fungi via the micro-dilution broth method to evaluate the minimal inhibitory concentration (MIC) following the European Committee on Antimicrobial Susceptibility Testing (EUCAST). Specifically, they were tested against four Gram-positive bacteria: *Streptococcus mutans*, *S. viridans*, *Staphylococcus aureus* (ATCC 25923) and *Staphylococcus epidermidis* (ATCC 12228); four Gram-negative bacteria: *Escherichia coli* (ATCC 25922), *Enterobacter cloacae* (ATCC 13047), *Klebsiella pneumoniae* (ATCC 13883) and *Pseudomonas aeruginosa* (ATCC 227853); and three fungi: *Candida albicans* (ATCC 10231), *C. tropicalis* (ATCC 13801) and *C. glabrata* (ATCC 28838) in the ATCC (American Type Culture Collection). The minimal inhibitory concentrations (MICs) of the tested extracts were evaluated using the broth micro-dilution method. Sterile 96-well polystyrene microtitrate plates were prepared by dispensing 100 μL per well of the appropriate dilution of the tested essential oils in a broth medium, always with two-fold dilution, in order to obtain the final concentrations of the tested oils that ranged from 0.50 to 10 mg/mL. The inoculums that were prepared with fresh microbial cultures in sterile 0.85% NaCl to match the turbidity of the 0.5 McFarland standard were added to the wells to obtain a final density of 1.5 × 10^6^ CFU/mL for bacteria and 5 × 10^4^ CFU/mL for yeasts (CFU: colony forming units). After incubation (37 °C for 24 h), the MICs were assessed visually for the lowest concentration of the oils, showing the complete growth inhibition of the reference microbial strains. An appropriate DMSO control (at a final concentration of 10%), a positive control (containing the inoculum without the tested samples) and a negative control (containing the tested derivatives without the inoculum) were included on each microplate. Standard antibiotic netilmicin (at concentrations of 4–88 μg/mL) was used to control the sensitivity of the tested bacteria and sanguinarine for oral bacteria, whilst 5-flucytocine and itraconazole (at concentrations of 0.5–25 μg/mL), as well as amphotericin B, were used as controls against the tested fungi (SanofiParis, France, Diagnostics Pasteur, at concentrations of 30, 15, and 10 μg/mL). For each experiment, the pure solvent used was also applied as a blind control. The experiments in all cases were repeated three times and the results were expressed as mean values [34].

## 3. Results and Discussion

### 3.1. Isolation of the Essential Oils

All essential oils had a pale yellow or yellow color and a strong, characteristic odor. A relatively high yield was observed (Table 2), especially from the leaves of *J. sabina*, with the largest volume of oil (4.2 mL of essential oil/100g of dry weight). In contrast, *J. macrocarpa* was the poorest source of essential oil (yield 0.2 and 0.6 mL/100g for leaves and cones, respectively). In general, cones appeared to have a higher essential oil yield compared with the leaves, as is also described in the literature [37,38]. It is also noteworthy that the *Juniperus* species with scale leaves (*J. excelsa*, *J. turbinata*, *J. foetidissima*) were observed to have a better yield of leaf essential oil compared with those with needle leaves (*J. communis*, *J. macrocarpa*, *J. oxycedrus*, and *J. drupacea*) [39]. The species of *J. sabina* was characterized by leaf dimorphism, having two different distinct leaf types, which are affected by the degree of maturation in two forms: young leaves are needle-shaped with a diameter of 10–5 mm and adult mature leaves of 1–2 mm in diameter and scale-shaped [40]. In the present study, the leaves of *J. sabina* were scale-shaped.

### 3.2. GC-MS Analysis

The chemical composition of the studied essential oils was investigated using GC-MS (Appendix A). The main chemical categories of the identified compounds are presented in Table 2 and Figure 1 and the abundant components of the studied essential oils are presented in Table 3.

The essential oils of the studied *Juniperus* species presented a multitude of volatile components, namely, 41 to 64 identified components, which were terpenes and their derivatives. The identified components accounted for a percentage of 80.0–99.9%.

The essential oils of the leaves and cones had monoterpenes as the main class of metabolites, except for *J. oxycedrus* and *J. macrocarpa*, where sesquiterpenes or their oxygenated derivatives predominated. The percentage of monoterpenes in the leaves ranged from 12 to 59%, while it ranged from 15 to 74% in the cones.

The second main classes of substances were sesquiterpenes and oxygenated sesquiterpenes. The percentage of sesquiterpenes ranged from 8.8 to 50.1% in the leaves and 9.5 to 39.7% in the cones, with the highest amounts found in the species *J. communis*, *J. turbinata* and *J. macrocarpa*. The percentages of their oxygenated derivatives were 3.8–32.8% in the leaves and 1.0–24.8% in the cones, with the highest amounts appearing in the leaves of the species *J. excelsa*, *J. turbinata* and *J. macrocarpa*.

As shown in Table 3, α-pinene was the component in greater abundance in the essential oils from the cones, with percentages of 12.1–43.1%, while a large percentage was also observed in the leaves (5.7–26.6%), which agrees with other scientific studies [16,41,42,43,44,45,46,47,48]. In the present study, larger quantities appeared in the cones compared with the leaves of the same *Juniperus* species. From the experimental results, the highest percentage of α-pinene appeared in the essential oil of the cones of *J. turbinata* (43.1%). This percentage is comparable with previous studies, where in the essential oil of the *J. turbinata* cones, α-pinene was found at a percentage of 22.1% in a sample from Sounio (Attica) [44], while it was found at a percentage of 66.1% in samples from Marathon (Attica) [49].

In contrast, in the essential oils from the leaves of *J. excelsa*, *J. foetidissima* and *J. macrocarpa*, the main component was α-cedrol, with percentages of 22.4%, 27.7% and 12.8%, respectively, while it was detected in traces in *J. turbinata* species, which is confirmed by the literature [44,45]. In the species *J. drupacea* and *J. sabina*, α-cedrol was not detected. The absence of α-cedrol is confirmed by previous studies in *J. drupacea* [44,49], while for *J. sabina*, a study [50] showed its presence only in some populations in Italy, where the species were further classified in the variety *sabina*. In those, the amount of α-cedrol reached the percentage of 15.9%. A large percentage of α-cedrol was also observed in the essential oils of leaves of *J. excelsa* originating from the Balkan region [14,51]. In the essential oil of *J. excelsa* leaves from an area southwest of Mikri Prespa, α-cedrol was found at a percentage of about 22% [14], while in another study with a sample from the Prespa area, the percentage was 28% [51], which is in full agreement with the experimental results in the current study.

In the species *J. oxycedrus*, the most abundant component was caryophyllene oxide (13.6%), which appeared in small percentages in the rest of the studied species. This can be explained by the greater proportion in leaves compared with cones, as researchers from Turkey quantified this component at about 30% in the essential oil of leaves and about 4% in the essential oil of cones [47].

*J. sabina* is a juniper species example with a chemotype that is characterized by the presence of sabinene, as well as its esteric derivative trans sabinyl acetate. According to [50], depending on the ratio of the two metabolites in the leaf essential oil of the species, the chemotype is characterized as high sabinene or high *trans*-sabinyl acetate. In the present study, the amount of sabinene was found with a percentage of 36.1% in the leaves and 36.9% in the cones, while *trans*-sabinyl acetate, was found only in small percentages (0.1 and 0.2% in the leaves and cones, respectively). Essential oils with a high concentration of sabinene in leaves were also studied by researchers from Iran, where several studies have taken place [15,25,52], and in samples from Greece (42.5%, 59.7%), Turkey (56.1%), Bulgaria (41.2%) and Italy (28.4–40.2%) [50]. These last populations were further classified, in the variety *balkanensis*. However, neither the variety nor geographical region excludes the presence of individual chemotypes with high trans-sabinyl acetate [50]. Sabinene was also the second most abundant component in *J. communis* for the essential oil of leaves and cones (13.5% and 10.1%, respectively). This result agrees with a previous study from Germany, where it was found at a percentage of approximately 11% in this species [53], as well as with a study from other mountains in Greece, where it was detected at 0.27–16.47% [43]. This compound was also found in the essential oil of *J. excelsa* leaves at a percentage of 6.5%.

In addition, it was observed that essential oils from cones are richer in β-pinene and myrcene compared with the leaf essential oils. This was confirmed by a previous study, where 4.9% and 1.95% of *J. foetidissima* samples from Mt Parnassos (Central Greece) and 53.85% and 1.45% of *J. oxycedrus* samples from the area of Marathon (Attica region) were detected for cones and leaves, respectively [49]. Also, the experimental data agree with the literature, which states that the essential oil of *J. turbinata* is the richest in myrcene, with a percentage of about 4% in a sample from Corsica, France [46], as well as the essential oil of *J. excelsa*, with a percentage of about 2% in a sample from Turkey [16].

Another main component that was identified was limonene, which was found in higher percentages (30% and 32% in the leaves and cones, respectively) in the essential oil of *J. drupacea*. This content is in agreement with the amount of limonene (27%) that was reported in a previous study [44], where the samples were also collected from other regions in Greece, while in other studies, where samples from Greece, as well as from eastern locations (Turkey, Crimea), were included, limonene was found to occur at even higher percentages (49% and 62% in cones and leaves, respectively [49]; 55.2% in aerial parts [54]; and 46–55% in leaves [55]). Regarding this volatile metabolite, *J. drupacea* was followed by the species *J. excelsa* and *J. foetidissima* (about 7.5–11%). In previous studies, *J. excelsa* leaf essential oil from the Prespa area presented limonene at a percentage of 22.7% [51], while in a study of cones from Turkey, it was 1.44% [16].

The component germacrene D was found in abundance in the essential oils of *J. drupacea* (9.8% and 10.1% in the leaves and cones, respectively), while a study conducted in cones of this species found a comparable percentage (6.0%) [49]. Other species with high amounts of germacrene D were *J. turbinata* (with a percentage of 12.3% in leaves) and *J. macrocarpa* (with a percentage of 8.8% in cones), the last of which is confirmed by literature data, where samples from Corsica presented percentages of about 4–10% [46] and samples from Turkey had a percentage of about 14% [48].

Regarding the metabolite terpine-4-ol, the highest percentage appeared in the leaves and cones of *J. sabina* (6.8% and 7.5% in the leaves and cones, respectively). This component was also found in *J. sabina* var. *balkanensis* growing in Greece (3.9%, 4.4%) and in var. *sabina* growing in Spain (7.2%) in similar amounts [50], as well as in samples from Iran and Algeria, with percentages between 5.9 and 9.9% for leaves [15,56] and 4.3% for cones [15]. Additionally, terpine-4-ol was mentioned in a study of samples of *J. communis* from Turkey in percentages between 3% and 7% [47], which matches our results (6.3%) in the essential oil of leaves from this species.

A difference observed compared with the literature was the small percentage (less than 2.5%) that *J. excelsa* presented for thujone, as it was expected in a percentage close to 11% based on a previous study from Turkey [47]. Additionally, in *J. sabina* var *balkanensis*, α-thujone was found in essential oils of leaf samples from Bulgaria (12.4%) and central Italy (8.0%), while β-thujone was found in leaves from different collection sites in Bulgaria (12.4%) and Italy (8.4%) as well [50]. It is worth noting that the samples from Greece in the present study and previously did not contain either α-thujone or β-thujone, which are very toxic compounds and considered psychotropic neurotoxins, with the α isomer showing greater toxicity than the mixture of both isomers. Consequently, there are strict rules for the acceptable amount contained in food and drinks [57]. According to the EMA, the maximum acceptable dose in humans is 5 mg as a maximum intake over two weeks, while the upper limit of thujone in alcoholic beverages is 10 mg/kg (except in the case of production from Artemisia species, where the limit is 35 mg/kg) [58]. In the same matter, the toxic esteric monoterpene sabinyl acetate that was found to be the main compound in some essential oils of leaves of *J. sabina* [50,56,59], was detected only in traces (0.1% and 0.2% in essential oils of the leaves and cones, respectively) in our analysis. According to literature and clinical studies, sabinyl acetate has an abortive effect in the early stages of pregnancy, while in the latter stages, it can cause severe embryotoxicity [60]. According to [50], species classified in the same variety, as well as growing in the same region, can produce different essential oils with high sabinene or high sabinyl acetate chemotypes. Since, there are only two studies referring to the essential oil from leaves ([50], present study) and only the present study referred to the essential oil from cones, the species must be further examined to conclude whether there are individuals with a high sabinyl acetate chemotype in the country.

Moreover, the content of α-terpineol, which was found to be very low (less than 1.5%) in the studied species, was reported with a percentage of approximately 27% in the essential oil of *J. communis* cones in samples from Corsica and Tunisia [41,46]. In addition, a difference was observed in the component 4-epi-abietal, which was found in a sample of *J. turbinata* cones from Sounio in a proportion of 13.2% [44], while in the present study, it was less than 1%. Another difference was that manool oxide was found at approximately 14.7% in essential oil from *J. turbinata* cones from Marathon [49], while in the present study, it was quantified at less than 2%. In addition, the data for the citronellol differed, as it was calculated to have a percentage of less than 1%, while in samples of *J. communis* from three mountains in Greece, it was found to have proportions of approximately 5–15% [43].

The differences between our analysis results and the literature data can be attributed, among others, to the differences in the environmental conditions of the collection localities, as well as the time and the season of collection [61,62]. Studies also showed the effect of elevation on the composition of essential oils. In particular, the higher the elevation, the more the amount of oil increases and it becomes richer in chemical components between populations of the same species [39].

### 3.3. Antimicrobial Activity

All the essential oils were evaluated for their antimicrobial activity via the dilution method against eight Gram-negative and -positive bacterial strains and three human-pathogenic fungi (Table 4). According to Aligiannis et al. (2001) [63], who proposed a classification of plant materials based on MIC results (strong inhibitors: MIC up to 500 μg/mL, moderate inhibitors: MIC between 600 and 1500 μg/mL, weak inhibitors: MIC above 1600 μg/mL), all essential oils exhibited strong-to-moderate antimicrobial activity, while some essential oils (Jf-C, Jcc-C, Jt-L and Jt-C) exhibited extremely strong activity (MIC up to 90 μg/mL) against Gram-positive strains and one yeast strain (Jt-L and Jt-C). A stronger activity was observed in the essential oils of the cones and especially against Gram-positive strains of *Staphylococcus* and *Streptococcus* compared with the leaves, which is consistent with previous results [15,64,65]. In particular, the essential oils of cones of *J. foetidissima*, *J. communis*, *J. drupacea* and *J. turbinata* showed a better effect, while the essential oils of leaves or cones of *J. macrocarpa*, *J. sabina* and *J. excelsa* did not show strong antibacterial activity. From previous studies, we expected a moderate-to-high antibacterial activity of the *J. communis* cones against *Staphylococcus* strains [66], as well as for the species *J. excelsa* with a MIC of about 0.13 mg/mL [67]. The differences were in proportion to the different chemical compositions that appeared in the present study.

On the other hand, the essential oils of the cones of *Juniperus* species showed a stronger antifungal activity compared with the leaves, while they had a greater inhibition against *C. glabrata*. In particular, the essential oil of *J. turbinata* cones had the lowest MIC value against all strains studied, followed by a small difference with that of *J. communis*, which agrees with a previous study on *J. communis* cones samples from Bulgaria that had an inhibition zone of 27.8 mm [17]. In contrast, the essential oil of *J. foetidissima* leaves showed weaker antifungal activity, which was reported in the literature [15], as well as the essential oil of the aerial parts of *J. oxycedrus*, probably due to the greater proportion of essential oil in the leaves.

## 4. Conclusions

In the present study, 15 essential oils of all eight *Juniperus* species growing wild in Greece were studied and compared. Specifically, four samples, namely, *J. oxycedrus* subsp. *deltoides*, *J. excelsa*, *J. foetidissima* and *J. communis* subsp. *communis*, were collected from the area of Lake Prespa (Western Macedonia); two samples of *J. macrocarpa* and *J. turbinata* from the Attica region; *J. drupacea* from Peloponnesus; and the crude material of *J. sabina* from Mt Tzena, North Greece. The essential oils of *J. macrocarpa*, *J. excelsa* and *J. sabina* cones, as well as *J. communis* leaves from Greece, were studied for the first time. The studied essential oils presented a variety of volatile components, with the main chemical category being monoterpenes, except for *J. macrocarpa* and *J. oxycedrus* subsp. *deltoides*, which had a higher percentage of sesquiterpenes or their oxygenated derivatives.

Additionally, the antimicrobial activities of the essential oils of the Greek juniper species were studied. The essential oils of the studied cones compared with the leaves exerted stronger antibacterial activity, especially against Gram-positive bacterial strains of *Staphylococcus* and *Streptococcus* strains. In particular, the essential oils of *J. foetidissima*, *J. communis* and *J. turbinata* cones showed a stronger effect. Also, the essential oils of the cones showed better antifungal activity compared with the ones from the leaves.

The current study, in addition to reporting on not previously studied essential oils from Greece, led to the comparison between the volatiles from different species and geographic areas in the country. The outcome found a low content of thujone, and thus, lower toxicity in general, giving the opportunity for potential exploitation after appropriate further analyses and assays on the efficacy and safety of such essential oils, either in the direction of food market and/or healthcare industry.

## Figures and Tables

**Figure 1 foods-12-03506-f001:**
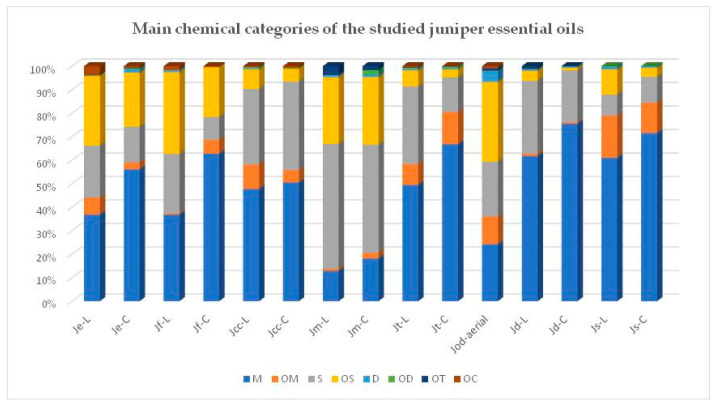
The main chemical categories of the studied juniper essential oils.

**Table 1 foods-12-03506-t001:** Information on the samples’ collection.

Taxon	Abbreviation	Date	Location	Elevation (m)
*Juniperus excelsa* M. Bieb.	*Je*	29 September 2021	Mt Devas, *J. excelsa* woodland. Prespa National Park, NW Greece.	1000
*Juniperus foetidissima* Wild.	*Jf*	29 September 2021	Mt Devas, *J. excelsa* woodland. Prespa National Park, NW Greece.	1104
*Juniperus communis* L. subsp. *communis*	*Jcc*	29 September 2021	Mt Varnous, open shrubland dominated by *J. communis.* Prespa National Park, NW Greece.	1000
*Juniperus macrocarpa* Sibth. & Sm.	*Jm*	21 November 2021	Few individuals Zoumperi (Nea Makri). Attica, Greece.	10
*Juniperus turbinata* Guss.	*Jt*	22 November 2021	Mt Hymettus, open shrubland on limestone. Attica, Greece.	350
*Juniperus oxycedrus* L. subsp. *deltoides* (R.P. Adams) N.G. Passal. In Bernardo, Passalacqua & Peruzzi	*Jod*	22 May 2021	Sandy grassland. Prespa National Park, NW Greece.	853
*Juniperus drupacea* Labill.	*Jd*	3 November 2022	Mt Parnonas, open shrubland on limestone. SE Peloponnesus, Greece.	1540
*Juniperus sabina* L.	*Js*	22 October 2022	Mt Tzena, open shrubland above the treeline of *Fagus sylvatica*. Northwest Macedonia, Greece.	1400

**Table 2 foods-12-03506-t002:** The main chemical categories (relative concentration %) and yield (% *v*/*w*) of the studied juniper essential oils.

	*Je-L*	*Je-C*	*Jf-L*	*Jf-C*	*Jcc-L*	*Jcc-C*	*Jm-L*	*Jm-C*	*Jt-L*	*Jt-C*	*Jod-aerial*	*Jd-L*	*Jd-C*	*Js-L*	*Js-C*
EO yield (%, *v*/*w*)	1.1	1.0	1.3	2.0	0.8	2.5	0.2	0.6	0.8	3.0	0.5	1.4	0.9	4.2	3.0
Monoterpenes	34.0	54.4	34.7	60.9	41.7	47.5	12.2	15.7	46.2	60.8	20.5	56.0	74.5	59.4	71.5
Oxygenated Monoterpenes	6.7	2.8	0.5	5.8	9.1	5.0	0.9	2.1	8.3	12.5	10.0	0.9	0.6	17.4	12.9
Sesquiterpenes	20.4	14.7	23.9	9.5	28.1	35.5	50.1	39.7	30.9	13.4	19.8	28.1	21.8	8.8	11.0
Oxygenated Sesquiterpenes	27.3	22.3	32.8	20.5	7.2	5.0	26.7	24.8	6.3	3.0	28.7	3.8	1.0	10.4	3.8
Diterpenes	0.2	1.2	0.8	nq	0.4	nq	0.9	0.9	0.6	0.4	3.9	0.5	0.5	1.0	0.6
Oxygenated Diterpenes	0.1	0.6	0.1	nq	0.4	0.4	nq	1.6	0.5	0.7	0.1	0.2	0.1	0.3	0.1
Oxygenated Triterpenes	0.1	0.3	nq	nq	nq	nq	3.6	1.5	0.2	0.1	1.0	1.1	0.1	nq	nq
Unknown compounds	5.3	1.8	3.5	1.4	10.8	4.6	4.4	12.5	5.5	7.3	12.9	9.0	0.8	1.6	nq
Other compounds	3.5	0.6	1.5	0.4	0.4	0.7	nq	nq	0.4	0.1	0.7	nq	nq	nq	nq
Total (%)	97.6	98.7	97.8	98.5	97.7	98.7	98.8	98.8	98.9	98.3	97.6	99.6	99.4	98.9	99.9

*Je—Juniperus excelsa*, *Jf—Juniperus foetidissima*, *Jcc—Juniperus communis* subsp. *communis*, *Jm—Juniperus macrocarpa*, *Jt—Juniperus turbinate*, *Jod—Juniperus oxycedrus* subsp. *deltoides*, *Jd—Juniperus drupacea*, *Js—Juniperus sabina*; L—leaves, C—cones, aerial—leaves and cones were not distinguished, EO—essential oil, nq—not quantified.

**Table 3 foods-12-03506-t003:** The main components (relative concentration ≥ 0.1%) of the studied juniper essential oils.

	*KI*	*Je-L*	*Je-C*	*Jf-L*	*Jf-C*	*Jcc-L*	*Jcc-C*	*Jm-L*	*Jm-C*	*Jt-L*	*Jt-C*	*Jod-aerial*	*Jd-L*	*Jd-C*	*Js-L*	*Js-C*
α-pinene	939	12.4	30.6	15.7	36.4	15.3	22.5	11.1	12.1	26.6	43.1	7.4	5.7	25.8	6.4	13.1
sabinene	977	6.5	nq	nq	nq	13.5	10.1	nq	nq	nq	nq	nq	nq	nq	36.1	36.9
β-pinene	980	nq	2.8	0.6	2.4	nq	nq	0.2	0.8	1.6	5.1	0.4	nq	1.5	nq	nq
myrcene	993	1.7	3.7	1.3	3.3	3.2	5.1	0.5	1.3	3.1	4.0	3.2	1.8	2.9	4.4	4.3
δ-3-carene	1014	nq	nq	5.9	5.3	nq	nq	nq	nq	9.2	1.6	nq	17.9	9.4	nq	nq
limonene	1028	11.0	9.9	9.1	7.6	nq	6.4	0.2	0.8	2.4	2.0	5.9	30.0	32.0	2.8	4.2
terpinene-4-ol	1174	2.2	0.2	0.1	0.7	6.3	1.7	nq	0.1	0.2	0.5	0.6	0.2	0.3	6.8	7.5
b-caryophyllene	1419	1.9	2.5	2.9	1.5	2.4	1.2	3.4	2.7	5.4	2.0	6.0	1.8	1.6	0.3	0.7
thujopsene	1431	0.9	0.7	1.5	0.6	nq	nq	6.5	2.7	nq	nq	nq	nq	nq	nq	nq
germacrene-D	1481	1.8	1.8	1.3	0.3	5.2	6.2	5.4	8.8	12.3	3.0	0.4	9.8	10.1	0.6	1.0
δ-Cadinene	1524	2.6	0.6	1.4	0.2	4.5	3.9	8.4	6.8	2.2	0.8	1.2	1.6	2.0	3.3	2.1
caryophyllene oxide	1582	nq	nq	nq	nq	0.3	0.2	1.8	1.3	1.0	1.3	13.6	0.6	0.1	nq	nq
α-cedrol	1601	22.4	19.5	27.7	17.8	0.2	0.2	12.8	14.1	0.1	0.2	0.8	nq	nq	nq	nq
humulene epoxide II	1602	nq	0.1	nq	0.2	0.5	0.4	nq	0.5	0.7	0.7	7.3	0.7	nq	nq	nq

*Je—Juniperus excelsa*, *Jf—Juniperus foetidissima*, *Jcc—Juniperus communis* subsp. *communis*, *Jm—Juniperus macrocarpa*, *Jt—Juniperus turbinate*, *Jod—Juniperus oxycedrus* subsp. *deltoides*, *Jd—Juniperus drupacea*, *Js—Juniperus sabina*; L—leaves, C—cones, aerial—leaves and cones were not distinguished, KI—Kovats Index, nq—not quantified.

**Table 4 foods-12-03506-t004:** Antimicrobial activity of essential oils measured in terms of the MIC (mg/mL).

Sample	*S. aureus*	*S. epidermidis*	*P. aeruginosa*	*K. pneumoniae*	*E. cloacae*	*E. coli*	*S. mutans*	*S. viridans*	*C. albicans*	*C. tropicalis*	*C. glabrata*
*Je-L*	0.52	0.48	0.88	0.90	0.98	0.87	0.50	0.45	0.88	0.77	0.22
*Je-C*	0.32	0.24	0.85	0.88	1.05	0.85	0.50	0.43	0.85	0.79	0.18
*Jf-L*	0.60	0.57	0.90	0.94	0.95	0.99	0.42	0.40	0.97	0.84	0.49
*Jf-C*	0.10	0.09	0.83	0.91	1.12	0.93	0.09	0.08	0.62	0.28	0.19
*Jcc-L*	0.12	0.10	0.97	0.96	1.02	0.95	0.12	0.28	0.75	0.65	0.62
*Jcc-C*	0.14	0.08	0.88	0.95	1.00	0.98	0.08	0.09	0.57	0.25	0.10
*Jm-L*	0.52	0.48	0.88	0.90	1.15	0.87	0.52	0.45	0.88	0.77	0.22
*Jm-C*	0.47	0.42	0.93	0.85	0.97	0.79	0.41	0.37	0.83	0.72	0.19
*Jt-L*	0.15	0.10	0.88	0.97	1.00	0.98	0.09	0.12	0.67	0.25	0.08
*Jt-C*	0.09	0.08	0.82	0.85	0.85	0.78	0.05	0.08	0.50	0.21	0.06
*Jod-aerial*	0.10	0.09	1.24	0.98	1.12	1.18	0.21	0.24	1.15	0.98	0.88
*Jd-L*	0.13	0.12	0.98	1.00	1.10	1.12	0.18	0.22	0.75	0.70	0.66
*Jd-C*	0.38	0.35	0.95	1.00	1.12	0.95	0.28	0.25	0.79	0.66	0.42
*Js-L*	0.55	0.50	0.94	0.98	1.04	1.15	0.49	0.48	0.90	0.87	0.59
*Js-C*	0.45	0.38	0.92	0.98	1.12	0.94	0.42	0.40	0.98	0.85	0.62
*Netilmicin*	4 × 10^−3^	4 × 10^−3^	8.8 × 10^−3^	8 × 10^−3^	8 × 10^−3^	10 × 10^−3^					
*Sanguinarine*							0.015	0.015			
*5-flucytocine*									0.12 × 10^−3^	1.25 × 10^−3^	10 × 10^−3^
*itraconazole*									1.3 × 10^−3^	0.7 × 10^−3^	0.65 × 10^−3^
*Amphotericin B*									0.9 × 10^−3^	0.6 × 10^−3^	0.4 × 10^−3^

*Je—Juniperus excelsa*, *Jf—Juniperus foetidissima*, *Jcc—Juniperus communis* subsp. *communis*, *Jm—Juniperus macrocarpa*, *Jt—Juniperus turbinate*, *Jod—Juniperus oxycedrus* subsp. *deltoides*, *Jd—Juniperus drupacea*, *Js—Juniperus sabina*; L—leaves, C—cones, aerial—leaves and cones were not distinguished.

## Data Availability

The data used to support the findings of this study can be made available by the corresponding author upon request.

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
