# Peer review of "Volatiles of All Native Juniperus Species Growing in Greece—Antimicrobial Properties"

_foods, 2023, doi:10.3390/foods12183506_

Round 1

Reviewer 1 Report (New Reviewer)

1. Line 139. Please check the Line Spacing. It is different from the rest of the article.

2. Line 161. Why did you use DMSO 10%? Any reference to it? Just for consideration, a higher concentration of DMSO used to dissolve the essential oil may hinder the result of antimicrobial activity.

3. Line 195, Table 3. Is the Kovats Index (KI) information in the table obtained from the literature? If yes, I suggest you calculate the KI again based on the interpolation of retention time and n-alkanes from your samples. Add this information to the table.  

4. Do you also use the standard EO compounds in the analysis of essential oil? If yes, you can add the information to your methodology.

5. Regarding the MIC, what is the basis for MIC ? It is MIC50, MIC80, MIC90? Due to its importance in the interpretation of data, Please add this information to your methodology. 

6. Line 331 Antimicrobial activity. Regarding the results, do you think your samples exhibited strong antimicrobial activity? I suggest you add the discussion in relation to the interpretation of the results. 

7. Line 369, The term "Gram" must be written in Capitals. 

8. Conclusion is too long. Paragraph 3 is written like in discussion part. Please revise it. 

The article must be proofread by a Native speaker due to grammatical errors found.

Author Response

  1. Line 139. Please check the Line Spacing. It is different from the rest of the article.

The paragraph has been edited accordingly.

  1. Line 161. Why did you use DMSO 10%? Any reference to it? Just for consideration, a higher concentration of DMSO used to dissolve the essential oil may hinder the result of antimicrobial activity.

It has been already noted in the methodology that “An appropriate DMSO control (at a final concentration of 10%)” It has never used in higher amounts than this.

In several methodologies for Antimicrobial activities for natural products such approach has been used. A selection of a couple of them, very recent and at mdpi journal are stated below:

-Widelski J, OkiÅ„czyc P, Paluch E, Mroczek T, Szperlik J, Å»uk M, Sroka Z, Sakipova Z, Chinou I, Skalicka-Woźniak K, Malm A and Korona-GÅ‚owniak I . The Antimicrobial Properties of Poplar and Aspen–Poplar Propolises and Their Active Components against Selected Microorganisms, including Helicobacter pylori. Pathogens. 2022; 11(2):191.

-Widelski, J.; OkiÅ„czyc, P.; SuÅ›niak, K.; Malm, A.; Paluch, E.; Sakipov, A.; Zhumashova, G.; Ibadullayeva, G.; Sakipova, Z.; Korona-Glowniak, I. Phytochemical Profile and Antimicrobial Potential of Propolis Samples from Kazakhstan. Molecules 2023, 28, 2984. https://doi.org/10.3390/molecules28072984

  1. Line 195, Table 3. Is the Kovats Index (KI) information in the table obtained from the literature? If yes, I suggest you calculate the KI again based on the interpolation of retention time and n-alkanes from your samples. Add this information to the table.  

The Kovats Index (KI) information have been calculated according to retention times of compounds and n-alkanes for our samples. The KI for all compounds are referred in the Supplementary and for the main compounds in Table 3.

  1. Do you also use the standard EO compounds in the analysis of essential oil? If yes, you can add the information to your methodology.

In the current analysis, standard EO compounds were not used for the identification of compounds from the examined samples. The n-alkanes were used in order to calculate KI for the validation   of the compounds.

  1. Regarding the MIC, what is the basis for MIC ? It is MIC50, MIC80, MIC90? Due to its importance in the interpretation of data. Please add this information to your methodology.

 There is already used in the antimicrobiological methodology the phrase” the MICs were assessed visually for the lowest concentration of the oils, showing the complete growth inhibition of the reference microbial strains”

  1. Line 331 Antimicrobial activity. Regarding the results, do you think your samples exhibited strong antimicrobial activity? I suggest you add the discussion in relation to the interpretation of the results. 

In this part of the discussion, we added a paragraph and the reference [63], with the classification of essential oils, as strong, moderate, or weak antimicrobial agents, based on MIC results ((strong inhibitors: MIC up to 500 μg/mL; moderate inhibitors: MIC between 600 and 1500 μg/mL; weak inhibitors: MIC above 1600 μg/mL).

Also standard antibiotics and antifungal agents with their MIC values for the examined strains were added in Table 4. These agents were used as reference compounds in order to determine the relative antimicrobial activity that EOs exhibited and further classify them as strong, moderate, or weak antimicrobial agents, according to the proposed classification as we described in the discussion.

  1. Line 369, The term "Gram" must be written in Capitals. 

The term has been corrected according to your recommendation.

  1. Conclusion is too long. Paragraph 3 is written like in discussion part. Please revise it. 

The third paragraph has been revised according to your suggestion.

Reviewer 2 Report (New Reviewer)

I believe that the corrected manuscript could be processed further. I think that the experimental part should be supplemented with data for the identification of washable components. The compounds were identified by mass spectra comparison with libraries (WileyRegistry of Mass Spectral Data or other) and confirmed by comparison of Kovats retention indices (KI) with literature data (Adams, 2001)…..

It should also be emphasized that these are relative concentrations.

After carefully looked at the submitted manuscript entitled: "Volatiles of all native Juniperus species growing in Greece - Antimicrobial properties". I still think that the corrected manuscript could be accepted for publication. The authors provide in great detail the metabolites of the mentioned plant grown in different locations in Greece. This is very important because the production of plant metabolites is very dependent on the geographical origin. The presence and ratio of certain metabolites can be significantly changed, and this produces a change in the biological activity of the extracts. In my opinion, the authors should improve that part a little and clearly state whether the identification was determined only on the basis of the Kovats index or whether they also used standards. If so, which ones. It should also be stated whether the quantification was done using appropriate standards or if it is a relative quantification. The conclusion contains enough argumentative details. The authors cite appropriate references. In Tables 2 and 3 as well as S2 in the empty fields, you should write not quantified. I still think that with these small additions, the submitted manuscript should be processed further in the journal.

Author Response

I believe that the corrected manuscript could be processed further. I think that the experimental part should be supplemented with data for the identification of washable components. The compounds were identified by mass spectra comparison with libraries (WileyRegistry of Mass Spectral Data or other) and confirmed by comparison of Kovats retention indices (KI) with literature data (Adams, 2001)…..

It should also be emphasized that these are relative concentrations.

The prosposed phrase regarding the identification of components has been added in the 2.3. GC-MS analysis part and the explanation of “relative concentration” has been added in Tables 2, 3 and S2.

Additional comments (Reviewer 2)

After carefully looked at the submitted manuscript entitled: "Volatiles of all native Juniperus species growing in Greece - Antimicrobial properties". I still think that the corrected manuscript could be accepted for publication. The authors provide in great detail the metabolites of the mentioned plant grown in different locations in Greece. This is very important because the production of plant metabolites is very dependent on the geographical origin. The presence and ratio of certain metabolites can be significantly changed, and this produces a change in the biological activity of the extracts. In my opinion, the authors should improve that part a little and clearly state whether the identification was determined only on the basis of the Kovats index or whether they also used standards. If so, which ones. It should also be stated whether the quantification was done using appropriate standards or if it is a relative quantification. The conclusion contains enough argumentative details. The authors cite appropriate references. In Tables 2 and 3 as well as S2 in the empty fields, you should write not quantified. I still think that with these small additions, the submitted manuscript should be processed further in the journal.

In the current analysis, standard compounds were not used for the identification of compounds from the examined samples. The compounds were identified by mass spectra comparison with libraries (Wiley) and confirmed by comparison of Kovats retention indices (KI) with literature data (Adams, 2001). The explanation that this is a “relative concentration” has been added in Tables 2, 3 and S2.

In Tables 2 and 3 as well as S2 in the empty fields, “nq” (= not quantified) has been added.

Reviewer 3 Report (New Reviewer)

In this study, the authors have examined the composition and antimicrobial activity of essential oils from 15 different species of Junipers located in Greece. The article is interesting and easy to read, although it has some shortcomings, such as poor editing, including multiple corrections, and the absence of statistical treatment of the data obtained. In this regard, it would be advisable to include some graphs with the most relevant data to enable a better visualization of the results.

With minor modifications, I believe that the article can be processed further in  Foods.

Author Response

Review Report (Reviewer 3)

In this study, the authors have examined the composition and antimicrobial activity of essential oils from 15 different species of Junipers located in Greece. The article is interesting and easy to read, although it has some shortcomings, such as poor editing, including multiple corrections, and the absence of statistical treatment of the data obtained. In this regard, it would be advisable to include some graphs with the most relevant data to enable a better visualization of the results.

With minor modifications, I believe that the article can be processed further in  Foods.

A graph with the main chemical categories of the studied juniper essential oils have been inserted as Figure 1 in the Results part of the manuscript.

This manuscript is a resubmission of an earlier submission. The following is a list of the peer review reports and author responses from that submission.

Round 1

Reviewer 1 Report

Ref.: foods-2509142-peer-review-v1

Title: Volatiles of all native Juniperus species growing in Greece - Antimicrobial properties

Authors: Fotiadou et al.

Reviewer’s comments for author(s)

Major editorial corrections are mandatory.

The goal of the study should be made clear.

Abstract. Check line 24 “thujonw???”

Introduction. Lines 48 to 52 require reference.

If not all, at least the European Pharmacopoeia should be cited in reference list.

It is not clear if just one sample, at one moment in time, was studied per species.

Table 2 is not necessary and the most important information of the table, the essential oil (EO) yield can be added to what is now Table 3., by adding and extra row following the table header, with EO yield (%, v/w).

Table 3. Consider using the code defined for each species in Table 1 plus a code for leaves (L) or cones (C). For instance, it would read “Jc C” for Juniperus communis L. subsp. communis. This would allow to write in regular orientation way (not in the vertical) and would be easier to read. The letter “C” or “L” could even come in a second line:

Jcc   Jcc

  C      L

Consider also to use in the now Table 3, the same alphabetic order of species as in Table 1.

Data can be presented with just one decimal figure an remove all “-“. If the cell is empty that is already because the compound was not detected.

Which triterpenes were found in the essential oils? Triterpenes are not volatile.

Retention index of each component should be added to the Table 4. Which was the percentage level to define “main components”?

Revise all document for lack of italics or symbols in components names.

Re-organize the results to address sequentially each of the species essential oil. Th results section already contains discussion, and either combine both, or separate results more clearly from discussion.

The discussion section repeats much of what is materials and methods and results and must be reformulated.

n/a

Author Response

Major editorial corrections are mandatory.

  • The goal of the study should be made clear.

The goal of the study was the phytochemical analyses of the essential oils from cones and leaves from all Juniperus growing wild in Greece. The evaluation of their quality in comparison with the results of other studies, as well as their potential applications, if possible, was among the intended scopes. As explained in the manuscript, in Greece there is a number of unique natural Juniperus forests, while several among them (species and subspecies) essential oils have not been previously studied. As Juniper essential oils have been used widely in food, cosmetic and/or medicinal industry it has been planned to test their preliminary bioactivities in vitro against a panel of common human pathogenic microorganisms as well, in order to evaluate their future applications, after appropriate further requested  studies on efficacy and safety.

  • Check line 24 “thujonw???”

The last two sentences have been rephrased.

  • Lines 48 to 52 require reference.

A ref has been added.

  • If not all, at least the European Pharmacopoeia should be cited in reference list.

The EMA is refereed as 9 and 10. The link address and the accessed date have been added.

  • It is not clear if just one sample, at one moment in time, was studied per species.

Yes, one sample, at one moment in time, was studied per species.

  • Table 2 is not necessary and the most important information of the table, the essential oil (EO) yield can be added to what is now Table 3., by adding and extra row following the table header, with EO yield (%, v/w).

Table 2 has been deleted and the yield has been added to the Table 3 (which has been renumbered as Table 2).

  • Table 3. Consider using the code defined for each species in Table 1 plus a code for leaves (L) or cones (C). For instance, it would read “Jc C” for Juniperus communis subsp. communis. This would allow to write in regular orientation way (not in the vertical) and would be easier to read. The letter “C” or “L” could even come in a second line:

Jcc   Jcc

  C      L

The suggested change has been adopted in the Table 2.

  • Consider also to use in the now Table 3, the same alphabetic order of species as in Table 1.

The same alphabetic order of species and the same code has been used in Table 3 as in the previous Tables.

  • Data can be presented with just one decimal figure an remove all “-“. If the cell is empty that is already because the compound was not detected.

The suggested changes have been adopted in the Tables 2 and 3.

  • Which triterpenes were found in the essential oils? Triterpenes are not volatile.

The oxygenated triterpenes that were detected in the essential oils of some species included in the study are shown in the table above.

Je-L

Je-C

Jm-L

Jm-C

Jt-L

Jt-C

Jod-aerial

Jd-L

Jd-C

Oxygenated Triterpenes

13-Epi-Manool Oxide,

Manool Oxide

Manool Oxide

13-Epi-Manool Oxide

13-Epi-Manool Oxide

Manool Oxide

13-Epi-Manool Oxide

13-Epi-Manool Oxide

Manool Oxide

Manool Oxide

13-Epi-Manool Oxide

% as shown in Table 2

0.1

0.3

3.6

1.5

0.2

0.1

1.0

1.1

0.1

Although triterpenes are molecules with high molecular weight, especially their oxygenated derivatives, their presence in the Juniperus essential oils is not uncommon. Previous studies conducted in the same species also confirm their presence in the oil of both leaves and cones. [Vourlioti-Arapi F., Michaelakis A., Evergetis E., Koliopoulos G., Haroutounian S. A., Essential oils of indigenous in Greece six Juniperus taxa. Parasitology Research 2011, 110, 5, 1829–1839, 51.54; Sezik E., Kocakulak E., Ozek T. Husnu Can Baser K.,Essential oil composition of Juniperus drupacea Lab. leaf from Turkey ,Acta Pharm Sci 2009, 51(2):109-120]

  • Retention index of each component should be added to the Table 4. Which was the percentage level to define “main components”?

The Kovats retention index (KI) have been added into the Table 4 and Table S2 (supplementary file).

The percentage level (≥0.1%), for the definition of the “main components”, has been added in the Table 3 title.

  • Revise all document for lack of italics or symbols in components names.

The whole document has been checked for Italics and symbols and they have been corrected accordingly.

  • Re-organize the results to address sequentially each of the species essential oil.

The results were organized according to the main metabolites that characterize each essential oil and were presented in Table 3. Our scientific group wanted to address the differences and similarities between the species and this seemed to be the best possible way as we mention the presence of every metabolite in all studied species. Each paragraph begins with the species that the specific metabolite was found in the greatest abundance and then we compare the percentages found in the other species included in the study as well as in studies from bibliography.

We hope the readers can follow up our way of thinking as the toxicity and antimicrobial activity should be connected to the chemical profile and the presence of specific metabolites in the essential oils. Data presented in this way, helps to address the reason why we suggest some species as more promising for use in the food and cosmetic industries than others, and this is the most used way of presentation of the results, according the existing literature.

  • The results section already contains discussion, and either combine both, or separate results more clearly from discussion. The discussion section repeats much of what is materials and methods and results and must be reformulated.

The Results and the Discussion parts have been combined and reformulated.

Reviewer 2 Report

This paper provides an interesting investigation into the chemical composition and antimicrobial resistance of essential oils derived from the leaves and cones of the eight Juniper species growing in Greece. Chemical composition was determined by GC-MS. Antimicrobial resistance was determined using broth microdilution to calculate the MIC. Details of the MIC method need to be provided, such as the broth, how many dilutions were assessed, and the increments between dilutions. The selection of microorganisms that were assessed needs to be justified, as it appears that the selection was based on what may have been available in the lab and not what was scientifically warranted. Details of the environment also need to be provided, such as the ambient temperature at which the leaves were dried, and the temperature and season during which the Juniper samples were obtained.

Please check the references as there is information missing and lack of italics in some references (journal names, scientific names). Italics for the scientific names are missing for the entire results section. Please check the entire manuscript. The results section contains portions which belong in a discussion. Either these need to be removed, or the results and discussion section can be combined. Please have the manuscript read and checked by a native English speaker. There are many grammatically incorrect sentences.

Abstract

Something happened with the last two sentences of the abstract.  Please fix this up. The last sentence requires rewording as it is not acceptable wording in English.

 Through antimicrobial tests, it was observed that the essential oils of the cones showed better activity compared to the leaves and especially the essential oils of J. foetidissima, J. communis and J. turbinata cones while appears safer in comparison with other oils towards the content in thujonw???. Taking into consideration the chemical profiles, safety and activities it could be clearly supported the potential use of Greek junipers for further exploitation in food industry.”

 Introduction

Line 32 please use the full word “approximately”.

Line 33 please change the slash to “or”, ie food/spice becomes “food or spice”

Lines 36-41 this requires references as it cannot all be attributed to general knowledge.

Line 46 Is Dutc supposed to be Dutch?

Lines 45-52 Please provide referencing.

Line 52 please end the sentence with a period.

Lines 56-57 Please explain the abbreviations.

Lines 56 and 400 & 401 These references are insufficient. Please provide full references.

Lines 60, 76, 77, 85 Please italicize scientific names here and throughout the document.

Line 98-99 This is not a complete sentence.

Material and Methods

Line 112 What is the environmental temperature, even approximately? Season?

Line 114 What is room temperature?

Line 131 Put a space between numbers and units.

Line 148 What was the broth medium? Did you follow a standard method, ie CLSI?

Line 151 Make sure that the power number is displayed in superscript, ie 1.5 x106 CFU/mL.

What is the rationale for the selection of these bacteria? They do not seem to be very diverse (4 Gm +ve cocci, comprising 2 species; 4 Gm -ve, of which 3 are coliforms; 3 fungi all of the same genus).  I hope it is more than just what happens to be in your culture collection.

How many MIC dilutions were tested? What was the increment between dilutions? MICs work by a doubling of each concentration, for example 1, 2, 4, 8, 16 etc. Were the microtiter wells dispensed in triplicate for each concentration (this is separate to the triplicate experiments)?

Table 2 spell out “ess”, ie essential?

Table 2 What are “aerial parts”? Can this be compared to the other data? 

Results

The scientific names are not in italics throughout the results section. Please italicize.

Line 275, 279 is a-thujone supposed to be α-thujone?

Line 283, 285 Put a space between numbers and units.

287-88 the compound was detected only in traces.  It cannot be “not detected” but then be in trace amounts.

Line 298 Please spell out approximately.

Line 309 Differences in season is why the season and time of year needs to be mentioned in the methods.

Line 331 The mention of zone inhibition in a previous, referenced study (Zheljazkov 2018) is not relevant since that study appeared to use a different method to determine MIC (zone inhibition on a plate, instead of broth microdilution). What might be relevant is if the species (J. communis) also had the lowest MIC compared to other species in that study.

Lines 362-363 Be careful about claims that antimicrobial activity was higher in the oils from the cones compared to the leaves. This was not always the case. Was there any statistical analysis performed to strengthen claims on activity from cones vs leaves?

Line 369 – previously has a spelling/typographic error

A lot of the information in the results that compares with other studies belongs in a discussion. Perhaps it would be better to combine the results and discussion. The journal allows for combined results and discussion.

References

Lines 417, 450 – Italicize journal name.

Line 422 – Number this reference.  How can it be found in the manuscript?

Line 459 Please provide book details.

Line 462 Italicize scientific name in the reference.

The quality of the English language is reasonable; however, there are insufficiently worded sentences on occasion. The entire manuscript would benefit from having a native English speaker read and double-check the grammar.

Author Response

This paper provides an interesting investigation into the chemical composition and antimicrobial resistance of essential oils derived from the leaves and cones of the eight Juniper species growing in Greece. Chemical composition was determined by GC-MS. Antimicrobial resistance was determined using broth microdilution to calculate the MIC.

Details of the MIC method need to be provided, such as the broth, how many dilutions were assessed, and the increments between dilutions.

The selection of microorganisms that were assessed needs to be justified, as it appears that the selection was based on what may have been available in the lab and not what was scientifically warranted.

Details of the environment also need to be provided, such as the ambient temperature at which the leaves were dried, and the temperature and season during which the Juniper samples were obtained.

Please check the references as there is information missing and lack of italics in some references (journal names, scientific names). Italics for the scientific names are missing for the entire results section. Please check the entire manuscript.

The results section contains portions which belong in a discussion. Either these need to be removed, or the results and discussion section can be combined. Please have the manuscript read and checked by a native English speaker. There are many grammatically incorrect sentences.

All the comments and suggestions have been considered in the revised version and the changes in details are the following:

Abstract

  • Something happened with the last two sentences of the abstract. Please fix this up. The last sentence requires rewording as it is not acceptable wording in English.

 “Through antimicrobial tests, it was observed that the essential oils of the cones showed better activity compared to the leaves and especially the essential oils of J. foetidissima, J. communis and J. turbinata cones while appears safer in comparison with other oils towards the content in thujonw???. Taking into consideration the chemical profiles, safety and activities it could be clearly supported the potential use of Greek junipers for further exploitation in food industry.”

The last two sentences have been rephrased accordingly.

Introduction

  • Line 32 please use the full word “approximately”.
  • Line 33 please change the slash to “or”, ie food/spice becomes “food or spice”
  • Lines 36-41 this requires references as it cannot all be attributed to general knowledge.
  • Line 46 Is Dutc supposed to be Dutch?
  • Lines 45-52 Please provide referencing.
  • Line 52 please end the sentence with a period.
  • Lines 56-57 Please explain the abbreviations.
  • Lines 56 and 400 & 401 These references are insufficient. Please provide full references.
  • Lines 60, 76, 77, 85 Please italicize scientific names here and throughout the document.
  • Line 98-99 This is not a complete sentence.

All the above corrections in the Introduction part have been made in the text following all corrections and suggestions.

Material and Methods

  • Line 112 What is the environmental temperature, even approximately? Season?

The season of plant collection was Autumn, apart from Juniperus oxycedrus L. subsp. deltoides which was collected at the end of Spring. The environmental temperature at the NW Greece in September is approximately at 200C and during May at 180C. At the region of Attica, the temperature during November is approx. 160C.

  • Line 114 What is room temperature?

Room temperature is 25°C and it has been added in the manuscript

  • Line 131 Put a space between numbers and units.

A space has been added between numbers and units throughout the paper.

Line 148 What was the broth medium? Did you follow a standard method, ie CLSI?

Apologies, as in different publications in MDPI  Journals, have been requested not to copy the methods used, in this case this is connected with the Antimicrobial assay, so probably some information towards the method used, have been omitted. All assayed essential oils were dissolved in dimethyl sulfoxide (DMSO) and screened for antibacterial and antifungal activities using the Mueller–Hinton broth, or RPMI with MOPS for the growth of fungi respectively. In all cases,  procedures followed the European Committee on Antimicrobial Susceptibility Testing (EUCAST) (www.eucast.org;).

  • Line 151 Make sure that the power number is displayed in superscript, ie 1.5 x106 CFU/mL.

Following your comment, it has been corrected.

  • What is the rationale for the selection of these bacteria? They do not seem to be very diverse (4 Gm +ve cocci, comprising 2 species; 4 Gm -ve, of which 3 are coliforms; 3 fungi all of the same genus). I hope it is more than just what happens to be in your culture collection.

With full respect to your surely strong background in microbiology, I guess from your comments focused to the microbiological assays, in this special case of phytochemical studies with potential antimicrobial activities, I can confirm you that this panel of microorganisms is among most widely selected ones for bioassays on herbal substances and extracts or essential oils, as in the present scientific work.

Of course, we have these microorganisms in our Lab, but not because we have selected them occasionally but on purpose and especially testing natural products, as phytochemistry is our scientific field of interest and to phytochemistry we are oriented since last decades. You can see in very recent publications with co-authors from Greece but also with colleagues from Microbiology Section (in Poland for example) very comparable panel of Gram positive, Gram negative and a number of Candida strains human pathogenic strains (Bozinou et al ,. Processes 11(2),394, 2023; Or by Widelski et al   Pathogens 11(2),191 2022 ).

Your comment that “the selection of the microorganisms in the in vitro assays was just among the ones in our Lab” make me and my co-authors feel uncomfortable, as we publish since last years, always evaluating the activities of natural products and have never received such comment before. Not fully understood what is the meaning of the phrase of the Reviewer “not what was scientifically warranted”. The strains are all of ATTC type, while have been used following comparable techniques, since last years. In any case we remain open to any further question for clarifications.

  • How many MIC dilutions were tested? What was the increment between dilutions? MICs work by a doubling of each concentration, for example 1, 2, 4, 8, 16 etc. Were the microtiter wells dispensed in triplicate for each concentration (this is separate to the triplicate experiments)?

Several standards dilutions were made followed the selected concentrations, and for the ones which showed strong activities (inhibition thereof), further dilutions manually performed. Yes, the microtiter wells dispensed in triplicate for each concentration used.

  • Table 2 spell out “ess”, ie essential?

The word has been corrected in Table 2.

  • Table 2 What are “aerial parts”? Can this be compared to the other data?

The explanation “Aerial parts: Leaves and cones were not distinguished” has been added at the end of Table 2.

Results

  • The scientific names are not in italics throughout the results section. Please italicize.

The scientific names have been corrected in Italics.

  • Line 275, 279 is a-thujone supposed to be α-thujone?

Yes, indeed and the name has been corrected accordingly.

  • Line 283, 285 Put a space between numbers and units.

Space has been added between numbers and units throughout the paper.

  • 287-88 the compound was detected only in traces. It cannot be “not detected” but then be in trace amounts.

The sentence has been corrected.

  • Line 298 Please spell out approximately.

The word approximately has been corrected.

  • Line 309 Differences in season is why the season and time of year needs to be mentioned in the methods.

The exact date of plant collection is already mentioned in Table 1. The average temperature during the period of plant collection (16-200C) has been added in the 2.1 Plant Material part of the manuscript.

  • Line 331 The mention of zone inhibition in a previous, referenced study (Zheljazkov 2018) is not relevant since that study appeared to use a different method to determine MIC (zone inhibition on a plate, instead of broth microdilution). What might be relevant is if the species ( communis) also had the lowest MIC compared to other species in that study.

As it is known there is no any special correlation factor or converter between different bioassays methods, but according to previous evaluation methods, the zones of inhibitions could be evaluated as inactive, weak, medium or strong activities. Of course, this approach is general and facilitate a bit the phytochemists working on these plants/species.

  • Lines 362-363 Be careful about claims that antimicrobial activity was higher in the oils from the cones compared to the leaves. This was not always the case. Was there any statistical analysis performed to strengthen claims on activity from cones vs leaves?

Actually, there is not any special statistical analysis, of course is not a general proposal on the essential oils bioactivities that the ones of cones are superior to leaves, but is only referred to presented results.

  • Line 369 – previously has a spelling/typographic error

The word has been corrected.

  • A lot of the information in the results that compares with other studies belongs in a discussion. Perhaps it would be better to combine the results and discussion. The journal allows for combined results and discussion.

The Results and the Discussion parts have been combined and some data were transferred to the Conclusion part following the Reviewer’s suggestions and proposals.

References

  • Lines 417, 450 – Italicize journal name.

The journal names have been italicized.

  • Line 422 – Number this reference. How can it be found in the manuscript?

It is already number 21. The repetition was a typo mistake.

  • Line 459 Please provide book details.
  • Book details have been added in ref 35 [Strid A, Bergmeier E, Fotiadis G. Flora and Vegetation of the Prespa National Park, Greece. Bilingual Edition: Greek- English, Aage V. Jensen Charity Foundation, by the Prespa Preservation Society and Vlassi Bros Publications 2020. ΙSBN: 978-9-6030228-5-5]
  •  
  • Line 462 Italicize scientific name in the reference.

The scientific name has been italicized.